# Faricimab for the Treatment of Diabetic Macular Edema and Neovascular Age-Related Macular Degeneration

**DOI:** 10.3390/pharmaceutics15051413

**Published:** 2023-05-05

**Authors:** Lorenzo Ferro Desideri, Carlo Enrico Traverso, Massimo Nicolò, Marion R. Munk

**Affiliations:** 1IRCCS Ospedale Policlinico San Martino, University Eye Clinic of Genoa, 16132 Genoa, Italy; 2Department of Neurosciences, Rehabilitation, Ophthalmology, Genetics, Maternal and Child Health (DiNOGMI), University of Genoa, 16126 Genoa, Italy; 3Department of Ophthalmology, Inselspital, Bern University Hospital, University of Bern, 3012 Bern, Switzerland; 4Bern Photographic Reading Center, Inselspital, Bern University Hospital, University of Bern, 3012 Bern, Switzerland

**Keywords:** faricimab, anti-VEGF drugs, intravitreal injections, ang/tie, ang2, angiogenesis, wet age-related macular degeneration, diabetic macular edema

## Abstract

Nowadays; intravitreal anti-vascular endothelial growth factor (VEGF) drugs are considered the first-line therapeutic strategy for treating macular exudative diseases; including wet age-related macular degeneration (w-AMD) and diabetic macular edema (DME). Despite the important clinical achievements obtained by anti-VEGF drugs in the management of w-AMD and DME; some limits still remain; including high treatment burden; the presence of unsatisfactory results in a certain percentage of patients and long-term visual acuity decline due to complications such as macular atrophy and fibrosis. Targeting the angiopoietin/Tie (Ang/Tie) pathway beyond the VEGF pathway may be a possible therapeutic strategy; which may has the potential to solve some of the previous mentioned challenges. Faricimab is a new; bispecific antibody targeting both VEGF-A and the Ang-Tie/pathway. It was approved by FDA and; more recently; by EMA for treating w-AMD and DME. Results from phase III trials TENAYA and LUCERNE (w-AMD) and RHINE and YOSEMITE (DME) have shown the potential of faricimab to maintain clinical efficacy with more prolonged treatment regimens compared to aflibercept (12 or 16 weeks) with a a good safety profile.

## 1. Introduction

Currently, wet-aged related macular degeneration (w-AMD) and diabetic retinopathy (DR) with diabetic macular edema (DME) are among the most frequent causes of visual impairment in the adult population in the Western countries [1,2]. In fact, the global prevalence of w-AMD is expected to increase from around 196 million patients nowadays to 288 million in 2040 [3]. Similarly, the increasing incidence of diabetes has caused the consequential rise of DR-related visual loss by 64% and blindness by 27%, respectively [4].

The current mainstay for the treatment of w-AMD and DME is represented by intravitreal drugs targeting the vascular endothelial growth factor (VEGF), which plays a crucial role in the pathogenesis of these exudative macular disorders [5]; however, real-life studies with anti-VEGF agents for treating w-AMD and DME have frequently shown inferior results as compared with pivotal clinical trials, due to the need of frequent injections, challenges with optimal disease monitoring, high costs and suboptimal response in some patients [6,7,8]. To overcome the above-mentioned limits, several different strategies have been investigated for treating these exudative retinal disorders. This included the exploration of different treatment regimens, the modification of dosing regimens, the investigation of novel, longer-lasting anti-VEGF agents, the inhibition of not only VEGF-A but also other members of the VEGF family and, nonetheless, the identification of other responsible molecular pathways as potential target [9,10,11,12,13].

The angiopoietin (Ang) yrosine kinase endothelial receptors (Tie) pathway has shown encouraging results for the management of w-AMD and DME [14].

In adults, the Ang/Tie pathway is involved in the regulation of vascular homeostasis, the modulation of vascular permeability and in neoangiogenic and proinflammatory processes [15]. In detail, Tie-2 is a transmembrane receptor, which is selectively located on endothelial cells in blood vessels, and it acts as a binding site for angiopoietins 1 and 2 (Ang-1 and Ang-2) [16]. These latter molecules exert a different biological activity; Ang-1 is a full Tie-2 agonist, inducing its phosphorylation and the activation of a downstream signaling process, which causes a positive effect on the vascular stability and inhibits vessel permeability and leakage; in contrast, Ang-2 acts as a partial agonist/antagonist of Tie-2. Therefore, its binding to Tie-2 receptor prevents the activation of the entire pathway and leads to vascular leakage [17].

Under physiological conditions, Ang-1 is constitutively expressed at higher concentrations than Ang-2 [18]; however, during pathological conditions (due to inflammation, ischemia/hypoxia or hyperglycemia) preformed Ang-2 is released from Weibel–Palade bodies of endothelial cells, provoking an increased Ang-2/Ang-1 ratio and a disruptive effect on vascular stability [19]. This angiogenic switch, which is characterized by Ang-2 overexpression, promotes the full response of endothelial cells to VEGF [14,20].

Interestingly, preclinical studies highlighted that the simultaneous dual inhibition of Ang-2 and VEGF-A was superior to either anti–VEGF-A or anti–Ang-2 activity alone [21].

Thus, the novel bispecific antibody, faricimab (faricimab-svoa; Vabysmo™) targeting both VEGF-A and Ang-2, has been recently developed by Roche/Genentech for the treatment of w-AMD and DME [22]. After meeting the primary endpoints in the phase III clinical trials (TENAYA and LUCERNE for w-AMD and YOSEMITE and RHINE for DME), faricimab was approved in USA in January 2022 and in Japan two months later for the treatment of w-AMD and DME. Faricimab was recently approved (September 2022) also by EMA with the same indications of FDA [23].

In this review, we will discuss the pharmacological, safety and clinical efficacy profile of faricimab, and analyze its possible beneficial role in patients with w-AMD and DME.

## 2. Materials and Methods

A literature search was performed in order to find all the published studies dealing with pharmacodynamics, clinical efficacy, and safety of faricimab for treating w-AMD and DME from inception until February 2023. These following electronic databases were adopted: Medline, PubMed, Science Citation Index via Web of Science, and the Cochrane Library. The following search terms were used: ‘faricimab and ‘RG-7716’ alone or in combination with ‘age-related macular degeneration’, ‘diabetic macular edema’ ‘efficacy’, ‘safety’, ‘toxicity’. Current clinical trials on the efficacy and tolerability of faricimab were considered, by examining research registers such as http://www.cliniclatrials.gov/ (accessed on 14 April 2023).

### 2.1. Preclinical Pharmacology

Faricimab is a bispecific IgG1 antibody, which simultaneously binds and inhibits VEGF-A and Ang-2, respectively. Faricimab has an overall size of 150 kDa and its structure is formed by 2 antigen-binding fragments (Fabs), respectively binding to Ang-2 and VEGF-A, and a modified fragment crystallizable region (Fc region) [24]. In detail, in the fragment crystallizable region (FcyR) was eliminated through the introduction of p329G mutations, which aims to shorten the systemic half-life (t_1/2_) and to prevent the recruitment of immune cells. Moreover, the viscosity of the drug was reduced by introducing a Triple A mutation. The advantages of bispecific agents include the recruitment of effector cells, the reduction of systemic toxicities and modulation of internalization properties, and the presence of synergistic effects when cell-surface receptors are targeted [25].

Faricimab was developed by adopting a CrossMAB technology. This technology is based on the crossover of the antibody domain within one Fab-arm of a bispecific IgG antibody in order to enable correct chain association; this allows the heterodimerization of 2 different antigen binding domains in a single molecule; hence, this ‘cross-over’ process is associated with faricimab high affinity to both Ang-2 and VEGF-A, and nonetheless, to a better stability profile as opposed to natural antibodies [26].

Preclinical studies have largely reported the beneficial role of Ang-2 inhibition in promoting vascular stability and decreasing the sensitivity of blood vessels to the activity of VEGF-A [27]. Ischemic retinal models have highlighted that the concomitant expression of VEGF-A and Ang-2 were responsible for a greater progression of choroidal neovascularization (CNV) than the expression of VEGF-A alone. In particular, CNVs are sensitive to Ang2 and a high Ang2/VEGF ratio promotes regression, while high Ang2 in the setting of hypoxia and/or concomitantly high Ang2 and VEGF stimulate CNVs formation [28].

In another study, increased levels of both Ang-2 and VEGF-A were found in the vitreous of diabetic patients after vitrectomy, with Ang-2 concentration being a key marker of proliferative diabetic retinopathy (PDR) [29].

In murine models, the simultaneous inhibition of both Ang-2 and VEGF-A was proven to be effective against laser induced-CNV; the reduction of the vessel lesion number, permeability, neuron loss and an improvement in retinal edema was greater than with either agent alone [27]. Similarly, an in vivo study on mice models with experimentally induced CNV, revealed that the dual inhibition of Ang-2/VEGF-A was more effective in decreasing vascular leakage, immune reactivity and apoptosis than either agent alone [21].

Another preclinical study in mice with endotoxin-induced uveitis showed that the simultaneous inhibition of Ang-2 and VEGF-A induced a reduction in retinal leukocyte invasion and the suppression of NF-κB translocation; hence, this preclinical study suggested the possible role of faricimab also in inflammatory ocular diseases; however, it is already known that also the inhibition of VEGF-A, VEGF-R1, VEGF-R2 and PIGF displays some anti-inflammatory properties, including the maintenance of endothelium in the quiescent state and the preservation of the dynamic barrier function and anti-adhesion against leukocytes in circulation. Then, further preclinical studies are needed to better explain the molecular pathways involved in the anti-inflammatory activity by faricimab [30].

Another important feature of faricimab is its potential antifibrotic potency. An increased fibrotic response has been shown in diabetic moce, which were experimentally made deficient in Ang-1 gene expression. In glomerular cell fractions, an elevated ANGPT-2/ANGPT-1 ratio (which are the genes encoding for Ang-1 and Ang-2), was found in an ANGPT-1/2 conditional knockdown mouse model [31]. But further studies are warranted to provide additional evidence on the role of Ang/Tie pathway in preventing fibrosis in retinal diseases (Figure 1).

### 2.2. Pharmacokinetics Ad Metabolism

After intravitreal administration, faricimab has shown a C_max_ of 0.23 ± 0.07 and 0.22 ± 0.07 μg/mL in patients with w-AMD and DME, respectively. Furthermore, faricimab T_max_ is achieved after 2 days after the intravitreal injection. Following repeated intravitreal injections with a bimonthly regimen, the average plasma levels of faricimab are predicted to be 0.002–0.003 μg/mL. No accumulation of the drug has been observed in either the vitreous or in the plasma [32].

Furthermore, a phase I clinical trial referred that the plasma concentrations of faricimab raised proportionally up to the dose of 3 mg. Faricimab at 1.5 mg showed a 6.1-fold lower systemic exposure as opposed to bevacizumab at 1.25 mg [33].

Although the metabolic route has not been fully characterized, faricimab is thought to undergo catabolism in lysosomes to small peptides and amino acids similarly to endogenous IgG and it is excreted through the kidneys. Finally, its mean apparent systemic t_1/2_ is estimated to be 7.5 days, which is different from that of bevacizumab (18.7), ranibizumab (5.8 days) and aflibercept (11.4 days) [32].

### 2.3. Safety

In an open-label phase I study investigating the safety and tolerability profile of faricimab in patients with w-AMD, the bispecific antibody was well tolerated by the treated arm up to the dose of 6 mg. Ocular adverse events (AEs) occurred in 58.3% of the patients (*n* = 14/24), of which 13 patients experienced mild symptoms and only one of them severe ocular AEs, namely a transitory reduction in visual function, which spontaneously underwent self-resolution after 2 weeks. This case and another case of treatment withdrawal were not deemed to be related to the drug by the principal investigators. Furthermore, the most common drug-related mild-to-moderate AEs were conjunctival hemorrhage, mild eye swelling and pain, blepharitis, conjunctivitis, iridocyclitis and photophobia. Importantly, no severe systemic AEs were referred in the treated arm [33].

In the phase II AVENUE trial, 81.7% (*n* = 214/262) of the patients with w-AMD experienced at least 1 adverse event (AE) during the follow-up period; only one case of procedure-related retinal hemorrhage occurred in the group receiving monthly doses of 1.5 mg faricimab. Severe ocular AEs occurred in 5 patients. Additionally, 12.6% (*n* = 33/242) of the patients had systemic AEs that the principal investigators did not believe were caused by the study drug. In the monthly ranibizumab 0.5 mg, monthly faricimab 1.5 mg, monthly faricimab 6 mg, bimonthly faricimab 6 mg, and ranibizumab 0.5 mg/faricimab 6 mg sequential therapy groups, intraocular inflammation was recorded in 4.5%, 6.5%, 5.1%, 0%, and 3.1% of the patients, respectively [34].

The multicenter, phase II STAIRWAY trial investigated clinical efficacy and safety profile of faricimab in patients with w-AMD. Ocular AEs were recorded in 39.4% (*n* = 28/71) of the study subjects. In the monthly ranibizumab group, 81.3% of the patients (*n* = 13/16), in the 12-week faricimab group (*n* = 18/24), and in the 16-week faricimab group (*n* = 23/31), at least one adverse event (AE) was reported by 76.1% (*n* = 54/71) of the patients. No cases of endophthalmitis were documented.

Furthermore, in 60.5% of the patients (*n* = 43/71) occurred at least 1 non-ocular AE; however, none of them was deemed to be related to the compound [35].

In the multicenter, phase II BOULEVARD Study, no cases of severe ocular AEs (including endophthalmitis and intraocular inflammation) were described in the study sample of 224 patients with DME. 1.8% of the study participants (*n* = 4/224; 3 in the ranibizumab arm and *n* = 1/224 in the 6.0-mg faricimab arm) were reported to have vitreous hemorrhage. Moreover, 2.2% (*n* = 5/224) of the patients died during the follow-up period; none of these events, nevertheless, were attributable to the medication (two in the ranibizumab group, one in the 1.5-mg faricimab group, and two in the 6.0-mg faricimab group) [36].

In the large, multicenter, phase III TENAYA and LUCERNE trials, faricimab was well-tolerated. The overall percentage of patients discontinuing the therapy was low, with only 0.9% of the patients (*n* = 3) discontinuing faricimab in both groups in TENAYA and 2.4% (*n* = 8) in the faricimab arm and 0.3% (*n* = 1) in the aflibercept arm in LUCERNE. Moreover, ocular AEs had a similar incidence between faricimab and aflibercept groups in both TENAYA (faricimab *n* = 121 [36.3%] vs. aflibercept *n* = 128 [38.1%]) and LUCERNE (*n* = 133 [40.2%] vs. *n* = 118 [36.2%]. Likewise, common non ocular AEs was similar between the treatment groups and no safety concerns were raised. In this regard, the incidence of serious ocular AEs in the study eye was 1.2% (*n* = 4) in the faricimab group and 1.8% (*n* = 6) in the aflibercept group in TENAYA and 2.1% (*n* = 7) in both groups in LUCERNE. 

Furthermore, the incidence of intraocular inflammation was low in both trials for both faricimab (TENAYA *n* = 5 [1.5%], LUCERNE *n* = 8 [2.4%]) and aflibercept (TENAYA *n* = 2 [0.6%], LUCERNE *n* = 6 [1.8%]. Among them only 2 cases (1 uveitis and 1 vitritis) treated with faricimab had a visual loss greater than 30 letters; and only the 1 case with uveitis did not resolve completely at the time of primary analysis [37].

In the multicenter, phase III YOSEMITE and RHINE clinical trials, faricimab was proven to have an overall safe profile and it was well tolerated. The reported incidence of ocular AEs was comparable among all the treatment arms (in YOSEMITE *n* = 98 [31%] and RHINE *n* = 137 [43%] in the bimonthly faricimab group, in YOSEMITE *n* = 106 [34%] and RHINE *n* = 119 [37%] in the faricimab personalized treatment interval (PTI) group, and in YOSEMITE *n* = 102 [33%] and RHINE *n* = 113 [36%] in the bimonthly aflibercept group). Most of these ocular AEs were mild or moderate. Furthermore, the incidence of intraocular inflammation was low across both trials (in the bimonthly faricimab, YOSEMITE *n* = 5 [1.6%], RHINE *n* = 3 [0.9%] and in the faricimab PTI, *n* = 7 [2.2%], *n* = 2 [0.6%]) versus bimonthly aflibercept *n* = 3 [1.0%], *n* = 1 [0.3%]).

Most of the cases of intraocular inflammation were mild, apart from 3 cases treated with faricimab (2 cases of severe uveitis in the PTI group and 1 case of severe vitreitis in the bimonthly group) [38].

The ongoing, open-label, phase III trial Rhone-X is currently including DME patients previously included in YOSEMITE and RHINE, in order to analyze faricimab long-term safety and tolerability [39].

## 3. Clinical Trials

### 3.1. Pase I Studies

In the multicenter, open-label, phase I clinical trial, 24 patients with w-AMD refractory to 3 or more anti-VEGF injections in the previous 6 months and a best-corrected visual acuity (BCVA) of 20/40 to 20/400 (Snellen equivalent) were injected with faricimab administered at single and multiple doses.

The primary aim of the study was to assess the safety and tolerability of both single and multiple faricimab intravitreal doses. BCVA and central retinal thickness (CRT) changes were secondary outcomes. Faricimab was administered in a single dose during the single-dose phase at doses of 0.5 mg, 1.5 mg, 3 mg, and 6 mg, respectively. Six individuals were included in the multiple-dose phase and were given faricimab 3 mg and 6 mg.

Patients received a single intravitreal administration of faricimab in the single dose phase only at day 1 and several intravitreal administrations scheduled on days 1, 28, and 56 in multiple-dose components.

On day 28, the average BCVA gain from baseline in the single-dose group was reported to be +7 letters (with a range of 0 to +18 letters; *n* = 1), with 91% of patients (*n* = 10/11) experiencing a visual gain of three letters or more. A good BCVA outcome was still observed at day 56, with a median change of +6 letters from baseline (*n* = 10/11). 28 days after the third injection, the 3-mg multiple-dose group had a median BCVA change from baseline of 0.5 letters (range, 9 to +8 letters; *n* = 6), with 4 of the 6 patients maintaining baseline BCVA within 3 letters and 1 patient improving by 8 letters from baseline. The average BCVA deviation from baseline in the 6-mg multiple-doses group was +7.5 letters (range, +3 to +18 letters; *n* = 6) was described 28 days after the third administration, with 5 of 6 patients experiencing at least a +5-letter improvement from baseline.

The mean CRT reduction from baseline were −42 μm (range, −101 to 10 μm; *n* = 11) and −117 μm (range, −252 to −7 μm; *n* = 6) in the same groups, respectively [33].

The promising results supported further clinical evalution in phase II trials.

### 3.2. Phase II Studies

The Phase II AVENUE (NCT02484690) and STAIRWAY (NCT03038880) clinical trials were designed to study the efficacy of faricimab compared to ranibizumab in patients with w-AMD. The primary endpoints were the mean change in BCVA after 40 and 36 weeks in STAIRWAY and AVENUE trial, respectively. Both trials evaluated faricimab superiority over ranibizumab as primary endpoint [35,40].

Similarly, the BOULEVARD (NCT02699450) trial aimed to evaluate the efficacy of faricimab in patients with DME. BOULEVARD trial had a superiority design over ranibizumab, and the primary endpoint was the mean change in BCVA after 24 weeks [36].

#### 3.2.1. Phase II Studies in Patients with w-AMD

The phase II AVENUE trial was a multicenter trial with an active comparator–controlled and double-masked design. In this study, 263 patients with CNV secondary to w-AMD who had never received treatment were enrolled and randomly assigned to one of the following treatment groups: monthly ranibizumab 0.5 mg, monthly faricimab 1.5 mg, monthly faricimab 6 mg, bimonthly faricimab 6 mg, or sequential ranibizumab 0.5 mg/faricimab 6 mg therapy.

After 36 weeks, patients treated with monthly faricimab at the dose of 1.5 mg experienced a functional gain of +9.1 letters in BCVA, patients treated with faricimab at 6.0 mg every 4 and 8 weeks improved by +6.0 letters. Those treated with monthly faricimab at 6.0 mg showed a gain of +5.9 letters and, lastly, patients in the monthly ranibizumab group experienced a gain of +7.2 letters. Moreover, all the treatment groups experienced an anatomical improvement. The most significant reduction in CRT was found in the ranibizumab 0.5 mg/faricimab 6 mg sequential therapy arm (−185 μm).

In the AVENUE trial, faricimab did not meet the primary endpoint of superiority to monthly ranibizumab for the management of w-AMD; however, the promising anatomical and functional outcomes, together with the results of the STAIRWAY trial, led to further evaluation of faricimab in phase III clinical trials [40].

The phase II STAIRWAY was a multicenter, randomized and active comparator-controlled trial. Overall, 76 subjects with CNV secondary to w-AMD were randomized 1:2:2 to receive monthly intravitreal ranibizumab at 0.5 mg, or faricimab, 6.0 mg, administered every 12 or 16 weeks, respectively.

After 24 weeks, there was no evidence of disease activity in 65% (*n* = 36/55) of the patients who received faricimab. After 40 weeks, the monthly ranibizumab, 12-week faricimab, and 16-week faricimab groups’ respective mean BCVA improvements from baseline were +11.4 (80% CI, 7.8–15.0), +9.3 (80% CI, 6.4–12.3), and +12.5 (80% CI, 9.9–15.1). The average number of intravitreal injections after the 52-week follow-up period was 12.9, 6.7, and 6.2 in the groups receiving monthly ranibizumab, 12-week faricimab, and 16-week faricimab, respectively.

From an anatomical perspective, those patients treated with faricimab dosing every 12 and 16 weeks showed significant reductions in CRT and changes in total lesion area in FA comparable with those treated with monthly ranibizumab.

Hence, the management of patients with w-AMD with faricimab dosing every 12 and 16 weeks was proven to maintain the initial BCVA and anatomic improvements throughout the 52-week follow-up period and those outcomes were comparable with those obtained by monthly ranibizumab [35].

#### 3.2.2. Phase II Studies in Patients with DME

The multicenter, randomized, double-masked, phase II BOULEVARD trial recruited 229 patients with DME (168 treatment-naïve and 60 previously treated with anti-VEGF drugs). The subgroup of 60 patients previously treated, 13 of them (21.7%) received 1 single intravitreal injection, 9 of them (15.0%) 2–3 injections, 10 of them (16.7%) 4–9 injections, 4 of them (6.7%) 10 or more injections and 24 of them (40.0%) were not reported.

Patients were 1:1:1 to the following treatment groups: 6.0 mg faricimab, 1.5 mg faricimab, and 0.3 mg ranibizumab using a monthly treatment regimen. The subgroup of patients already treated with anti-VEGF drugs were assigned to either the 6.0 mg faricimab or the 0.3 mg ranibizumab groups.

The primary outcome was the change of BCVA from baseline at week 24.

Following the follow-up period, the 0.3 mg ranibizumab, 1.5 mg faricimab, and 6.0 mg faricimab treatment groups showed adjusted BCVA gains from baseline of +10.3 ETDRS letters (80% CI, 8.8–11.9 ETDRS letters), +11.7 ETDRS letters (80% CI, 10.1–13.3 ETDRS letters), and +13.9 ETDRS letters (80% CI, 12.2–15.6 ETDRS letters), respectively.

After 24 weeks, there were 59.2% (80% CI, 50.2–67.7%) of the patients in the ranibizumab group, 60.6% (80% CI, 51.6–68.9%) in the 1.5-mg faricimab group, and 72.1% (80% CI, 62.9–79.8%) in the 6.0-mg faricimab group with a gain of 10 letters or more from baseline.

In comparison to ranibizumab, faricimab was linked to a longer time to re-treatment interval over the follow-up period, a dose-dependent decrease in CRT, and improvements in the Diabetic Retinopathy Severity Score (DRSS).

In conclusion, faricimab led to a significant gain in BCVA as opposed to monthly ranibizumab in patients with DME after a follow-up period of 24 weeks [36].

### 3.3. Phase III Studies

#### 3.3.1. Phase III Studies in Patients with w-AMD

TENAYA (NCT03823287) and LUCERNE (NCT03823300) were 2 multicenter, randomized, double-masked, phase III clinical studies, which recruited together 1329 treatment-naïve patients with w-AMD older than 50 years old. The study subjects were randomized (1:1) to intravitreal faricimab 6.0 mg up to Q16W (every 16 weeks), based on protocol-defined disease activity criteria at weeks 20 and 24, or bimonthly aflibercept 2.0 mg.

The primary outcome was the average change in BCVA from baseline at weeks 40, 44, and 48 in the treated patients. Both trials were designed to evaluate faricimab non-inferiority to aflibercept as a primary endpoint.

The mean increase in BCVA over baseline was +5.8 letters (95% CI 4.6 to 7.1) for faricimab and +5.1 letters (3.9 to 6.4) for aflibercept, with a total numerical difference of +0.7 letters (−1.1 to 2.5) for TENAYA. With no numerical differences [−1.7 to 1.8] in LUCERNE, +6.6 letters [5.3 to 7.8] were recorded in the faricimab group and +6.6 letters [5.3 to 7.8] in the aflibercept group.

The mean modifications in CRT from baseline were −136.8 μm (95% CI −142.6 to −131.0) in the faricimab group and −129.4 μm (−135.2 to −123.5) in the aflibercept group in TENAYA (treatment difference −7.4 μm [−15.7 to 0.8]), and −137.1 μm (−143.1 to −131.2) in the faricimab group and −130.8 μm (−136.8 to −124.8) in the aflibercept group in LUCERNE (treatment difference −6.4 μm [−14.8 to 2.1]).

In conclusion, visual benefits with faricimab administered with a max Q16W regimen in patients with w-AMD were comparable and non-inferior with those obtained with bimonthly aflibercept, which indicates the potential to extend the treatment interval without losing efficacy [37].

#### 3.3.2. Phase III Studies in Patients with DME

YOSEMITE (NCT03622580) and RHINE (NCT03622593) were 2 multicenter, randomized, double-masked, phase III clinical studies, which enrolled 3247 patients (*n* = 1532 in YOSEMITE and *n* = 1715 in RHINE) with visual impairment due to center-involving DME. The following treatment regimens were given to study participants by randomization (1:1:1): faricimab 6.0 mg every 8 weeks, faricimab 6.0 mg per PTI, or aflibercept 2.0 mg every 8 weeks. Based on disease activity, the PTI group had a variable treatment schedule (every 4 weeks (Q4W) up to Q16W). The mean change in BCVA at 1 year, averaged throughout weeks 48, 52, and 56, served as the main endpoint. The primary aim of both trials was to examine faricimab non-inferiority to aflibercept.

The average BCVA improvements were +10.7 letters, +11.6 letters, and +10.1 letters in the case of YOSEMITE [treatment differences versus aflibercept, 0.2 letters (97.52% CI 2.0 to 1.6) and +0.7 letters (97.52% CI 1.1 to 2.5)] and +1.5 letters (97.52% CI 0.1 to 3.2) and +0.5 letters (97.52% CI 1.1 to 2.1) in RHINE [treatment differences vs. aflibercept] and +11.8 letters, +10.8 letters, and +10.3 letters in the bimonthly faricimab, faricimab PTI and bimonthly aflibercept groups, respectively. The reductions in CRT were significantly greater in patients treated with faricimab than those treated with aflibercept after 12 months. In YOSEMITE, average CRT variations from baseline were −206.6 μm (95.04% CI −214·7 to −198.4) in the bimonthly faricimab, −196·5 μm (−204·7 to −188·4) in the faricimab PTI and −170.3 μm (−178.5 to −162·2) in the aflibercept groups, respectively. Similarly in RHINE, average CRT decreased from baseline by −195·8 μm (−204·1 to −187·5) in the bimonthly faricimab, −187·6 μm (−195·8 to −179·5) in the PTI faricimab versus −170·1 μm (−178·3 to −161·8) in the aflibercept groups, respectively. After 52 weeks, in both YOSEMITE and RHINE, 52% of patients randomized to the faricimab PTI group remained in a Q16W interval and 21% of them in a 12-week (Q12W) regimen. Thus, faricimab showed strong durability in YOSEMITE and RHINE, considering that more than 70% of patients in the PTI groups maintained in a ≥Q12W dosing regimen at 1 year.

In conclusion, faricimab showed consistent visual and superior anatomical outcomes in patients with DME even in an extended the treatment regimen up to Q16W [38].

### 3.4. Other Ongoing Phase III and IV Trials

Currently, the ongoing, multicenter, randomized, phase III BALATON (NCT04740905) and COMINO (NCT04740931) trials are investigating the clinical efficacy of faricimab in comparison with aflibercept and sham for the management of branch retinal vein occlusion (BRVO) and central retinal vein occlusion (CRVO), respectively. Preliminary results showed that monthly faricimab was non-inferior to monthly aflibercept in terms of visual function up to 24 weeks [41,42].

In addition, the ongoing, multicenter, single-arm, phase IV ELEVATUM trial is studying the clinical response to faricimab in treatment-naïve patients with DME, which were underrepresented in the previous phase II and III trials [43].

The complete list of the clinical trials evaluating the clinical efficacy of faricimab for treating w-AMD, DME and BRVO/CRVO is summarized below (Table 1).

## 4. Discussion

Although anti-VEGF drugs still represent the mainstay for the treatment of exudative retinal disorders, including w-AMD and DME, other new therapeutic strategies are being examined with the aim to reduce the treatment burden and improve the efficacy and durability of intravitreal injections [6,10,45,46].

Currently, there are 4 intravitreal anti-VEGF agents approved by the USA FDA and EMA for treating w-AMD and DME: pegaptanib sodium (Macugen^®^, Montreal, QC, Canada), ranibizumab (Lucentis^®^ 2013 Genentech, South San Francisco, CA/Roche, Basel, Switzerland), aflibercept (AFL, Eylea^®^, Regeneron^®^, Tarrytown, NY, USA, Bayer^®^, Leverkusen, Germany) and only for w-AMD brolucizumab (Beovu^®^, Basel, Switzerland 2019); moreover, another anti-VEGF, bevacizumab, is largely adopted in clinical practice as an ‘off-label’ treatment option [1,47].

Real-life studies have often provided inferior clinical outcomes in comparison with large, multicenter pivotal trials [48,49]. This is partly attributed to the high treatment demand, which is often not sustainable in a busy daily clinic. It is therefore not surprising that drug durability is one of the most important attributes retina specialists are looking for in a new drug. The Ang/Tie pathway may represent another possible target in combination with VEGF, considering the augmented pathologic levels of Ang-2 in proinflammatory, hypoxic and proangiogenic conditions occurring in macular exudative diseases, leading to increased vascular permeability, leakage and enhanced sensitivity of endothelial cells to VEGF-A [20].

Faricimab (faricimab-svoa; VabysmoTM), a bispecific antibody with a molecular structure that binds to both VEGF-A and Ang-2 with high affinity, was recently licensed by the FDA and EMA [24]. According to preclinical findings, concurrent inhibition of VEGF-A and Ang-2 was more efficient than anti-VEGF monotherapy alone at lowering the number of CNV lesions, vascular leakage, and inflammation; however, despite the promising preclinical results of faricimab, some of the preclinical studies were funded by Roche and thus further preclinical evidence in this regard should be provided by spontaneous and non-sponsored studies [25].

The following large, multicenter, phase III clinical trials RHINE and YOSEMITE for DME and LUCERNE and TENAYA for w-AMD have provided evidence in a clinical setting that faricimab shows prolonged efficacy up to a Q16W treatment regimen, without being inferior to aflibercept in terms of visual and anatomical outcomes [37,38].

In terms of visual function, a fast and sustained improvement in BCVA was achieved with a faricimab dosing of either Q8W or PTI after 52 weeks, which was comparable to the visual function improvement in the Q8W aflibercept group. In the PTI groups, non-inferiority was obtained with fewer interval-determining visits as opposed to aflibercept [38].

In LUCERNE and TENAYA rapid BCVA gains were sustained up to week 48 and a comparable percentage of patients with w-AMD improved of 15 letters or more in both faricimab and aflibercept groups. In TENAYA 79.9% of the patients completed the 48-week follow-up period with a 12-week or 16-week regimen and similarly did 77.8% of the patients in LUCERNE. This high percentage reflect the capability of faricimab to prolong the treatment regimen without losing efficacy in a big part of the patients in comparison with aflibercept (8-week regimen) [37].

In RHINE and YOSEMITE a higher percentage of patients with DME showed absence of fluid in comparison with the aflibercept-treated subjects up to week 56 [38]. In this regard, it has been widely demonstrated that DME is a multifactorial disease, in which several pro-inflammatory cytokines and different processes are involved, and the bispecific activity of faricimab (targeting both Ang-2 and VEGF-A) may be particularly indicated [23]. Further studies should provide more evidence on the better anatomical outcomes shown in patients with DME rather than those with w-AMD.

Results from these large, multicenter trials suggest the potential of faricimab to achieve stable and durable results and the possibility to tailor a personalized treatment regimen in relation to individual needs, with important benefits in the reduction of the treatment burden and costs for both healthcare providers and patients. Further real-life clinical studies on faricimab should also assess the possible presence of gender, ethnic or age differences in the treated population; in fact, previous studies on anti-VEGF agents have reported the presence of gender and racial differences in the prevalence and disease severity in patients affected by DME and w-AMD [50].

Furthermore, it has been reported the negative effect of the dynamic fluctuations in CRT in patients with DME and w-AMD treated with anti-VEGF agents; in fact these dynamic changes are associated with poorer visual outcomes [51,52]. Faricimab has shown in the first real-life studies the ability to resolve the presence of intraretinal and subretinal fluid and stabilize CRT fluctuations in the subgroup of patients recalcitrant to other anti-VEGF agents, including aflibercept [53,54]. For this reason, this novel drug may represent an important weapon in the armamentarium of retina specialists.

Overall, faricimab has so far shown a good safety and tolerability profile, with an incidence of ocular (including intraocular inflammation) and systemic AEs comparable to that of aflibercept [23]. Also, no case of retinal occlusive vasculitis was reported. Similarly, also the combination therapy with nesvacumab revealed a good safety profile [55].

A potential limitation of phase III clinical trials is that they only compared faricimab with aflibercept; in the near future, further studies should provide evidence on the direct comparison between faricimab and the other approved anti-VEGFs. Moreover, results of the comparative analysis of faricimab show similar or non-inferior responses to aflibercept rather than showing superior therapeutic responses. Further real-life clinical studies should provide more evidence on the long-term anatomical and visual outcomes of faricimab in patients with w-AMD and DME.

This is in contrast with the recent FDA and EMA- approved brolucizumab (Beovu, Novartis™), a single-chain antibody fragment targeting VEGF-A. Although, brolucizumab has shown a great clinical efficacy in restoring BCVA and maintaining the absence of intraretinal and/or subretinal fluid in pivotal trials (HAWK and HARRIER) and in real-life studies, some safety concerns have been raised regarding the higher risk of intraocular inflammation and, in some cases, retinal vasculitis and retinal vascular occlusions [13,49]. In fact, in a large, real-life study on brolucizumab-treated patients, the Intelligent Research in Sight (IRIS) registry reported an incidence of intraocular inflammation and/or retinal vascular occlusions accounting for 2.1% of the study population [56]. To mitigate this risk, a recent consensus published by Holz et al. recommended a careful patient selection and education, an assessment of inflammation and intensive treatment with topical/systemic corticosteroids in case of inflammation after brolucizumab injection [57]. Overall, faricimab has shown a safer profile with lower intraocular inflammation rates compared to brolucizumab together with long durability up to Q16W, which could represent an important advantage in the clinical practice; however, further real-life clinical studies are warranted to prove faricimab safety, tolerability and durability in a real-life setting.

Another treatment modality targeting the Ang/Tie has been investigated for treating w-AMD. This consists of intravitreal nesvacumab plus aflibercept combination therapy a fixed dose providing specific doses of the anti-Ang2 antibody with a known effective dose of aflibercept at 2 mg. In the phase II RUBY (NCT02712008) trial, the primary endpoint was the mean change in BCVA after 12 weeks. The primary endpoint was to demonstrate the superiority of nesvacumab combination therapy than aflibercept alone. They reported better anatomical outcomes, but not visual benefit (which was the primary outcome) with combination therapy in comparison with aflibercept. Thus, since nesvacumab therapy did not show to be superior to aflibercept therapy, no phase III trials were continued [55].

Currently, together with faricimab, other novel drugs are being investigated for the management of w-AMD and DME [58,59]. KSI-301 is a novel anti-VEGF drug, characterized by an antibody biopolymer conjugate (ABC) structure with a molecular weight of 950 kDa [60]. These biomolecular properties have been reported to increase the ocular t _1/2_, which, according to the phase II GLEAM and GLIMMER clinical trials, may lead to an extension of the treatment regimen up to 16–20 weeks; however, recent partial results have shown that KSI-301 did not meet the primary endpoint of non-inferiority to aflibercept for treating w-AMD; however, further data are needed to provide more evidence on KSI-301 clinical efficacy [61].

Furthermore, the ranibizumab port delivery system (RPDS), which is a surgically-implanted device allowing the continuous delivery of ranibizumab into the vitreous for 6 months or more, has been approved for treating w-AMD and is under investigation in phase 3 trials for DME [62]; however, despite the promising and durable clinical outcomes, the benefit-risk balance of the need for surgical procedure (with the risk of vitreous hemorrhage) should be carefully considered [63].

Differently, abicipar pegol, a small DARPin protein targeting VEGF-A, despite the satisfactory clinical results has not been approved due to the high incidence of intraocular inflammation [11].

Currently, also gene therapy thorough the long-lasting VEGF blockage is attracting interest for the treatment of w-AMD; RGX-314, ADVM-022, and ADVM-032 are the most promising options in this regard [64]. In this respect, further, late-phase, clinical trials are needed to prove the clinical efficacy and safety of gene therapy for treating w-AMD.

Also the TGF-beta is being examined in early-phase clinical trials in order to prevent the fibrotic stage of macular edema in patients with w-AMD and DME [65]. Results in this respect are needed to better describe the potential role of TGF-beta as a biomolecular target in these patients.

Other early-phase studies are investigating the role of tyrosine kinase inhibitors, targeting VEGFR, PDGFR, fibroblast growth factor receptor (FGFR), and c-kit against CNVs. In this regard, a phase I study on oral X-82, targeting both VEGF and PDGF, showed in 35 patients with w-AMD that 60% of them did not need further anti-VEGF agents during the follow-up period. A side effect was the temporary elevation of liver enzymes [66]. Further advance-phase trials are warranted to provide evidence on oral tyrosine kinase inhibitors for treating w-AMD.

In conclusion, the modulation of Ang/Tie pathway with faricimab has proven to be a potential first-line treatment option in the management of w-AMD and DME, allowing to decrease the treatment burden and extend the regimen up to 16 weeks.

Further clinical trials are needed to better understand faricimab efficacy and safety in both treatment naïve-patients with DME and w-AMD and in those patients already treated with anti-VEGF agents, which were underrepresented in pivotal, phase III clinical trials (RHINE and YOSEMITE for DME and TENAYA and LUCERNE for w-AMD).

## Figures and Tables

**Figure 1 pharmaceutics-15-01413-f001:**
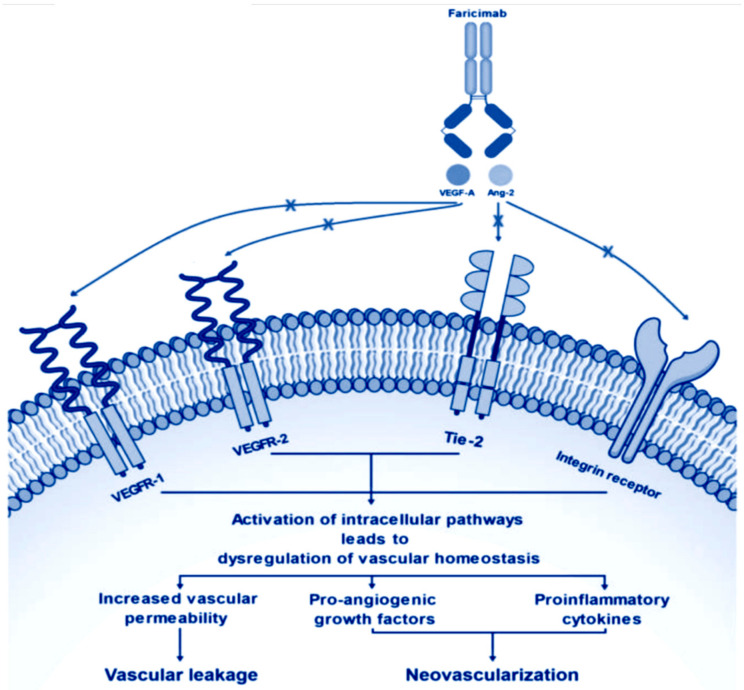
Molecular mechanism of action of faricimab leading to a decrease of vascular leakage and the inhibition of neovascular processes.

**Table 1 pharmaceutics-15-01413-t001:** Summary of the current, main clinical trials investigating faricimab clinical efficacy and safety in exudative macular diseases.

Trial Identifier	Phase	Indication	Number of Patients and Interventions	Results in BCVA	Status
AVENUE NCT02484690 [40]	II	w-AMD	263 patients; monthly ranibizumab 0.5 mg, monthly faricimab 1.5 mg, monthly faricimab 6 mg, bimonthly faricimab 6 mg, and ranibizumab 0.5 mg/faricimab 6 mg sequential therapy	At week 36, adjusted mean change in BCVA vs. ranibizumab was 1.6 (80% CI, −1.6 to 4.7) letters for faricimab 1.5 mg every 4 weeks (*p* = 0.52), −1.6 (80% CI, −4.9 to 1.7) letters for faricimab 6.0 mg every 4 weeks (*p* = 0.53), and −1.5 (80% CI, −4.6 to 1.6) letters for faricimab 6.0 mg every 8 weeks (*p* = 0.53). For faricimab 6.0 mg every 4 weeks adjusted mean change was −1.7 (80% CI, −3.8 to 0.4) letters (*p* = 0.30)	Completed
STAIRWAY NCT03038880 [35]	II	w-AMD	76 patients; monthly ranibizumab 0.5 mg, faricimab, 6.0 mg, every 12 or 16 weeks	At week 40, adjusted mean BCVA gains from baseline were +11.4 (80% CI, 7.8–15.0), +9.3 (80% CI, 6.4–12.3), and +12.5 (80% CI, 9.9–15.1) for the ranibizumab every 4 weeks, faricimab every 12 weeks, and faricimab every 16 weeks arms, respectively	Completed
LUCERNE NCT03823300 [37]	III	w-AMD	1012 patients; faricimab 6.0 mg up to every 16 weeks vs. bimonthly aflibercept 2.0 mg	Mean BCVA gain +6.6 letters [5.3 to 7.8] with faricimab 6.0 mg up to every 16 weeks and +6.6 letters with aflibercept 2.0 mg every 8 weeks [5.3 to 7.8]; treatment difference 0·0 letters [−1.7 to 1.8]	Completed
TENAYA NCT03823287 [37]	III	w-AMD	989 patients; faricimab 6.0 mg PTI up to every 16 weeks vs. bimonthly aflibercept 2.0 mg	Adjusted mean change + 5.8 letters [95% CI 4·6 to 7·1] with faricimab 6.0 mg up to every 16 weeks and + 5.1 letters [3.9 to 6.4] with aflibercept 2.0 mg every 8 weeks; treatment difference 0.7 letters [−1.1 to 2.5]	Completed
AVONELLE-X NCT04777201 [44]	III	w-AMD	1036 patients; faricimab 6.0 mg PTI up to 16 weeks	Awaiting results	Active, not recruiting
BOULEVARD NCT02699450 [36]	II	DME	229 patients; 6.0 mg faricimab, 1.5 mg faricimab, or 0.3 mg ranibizumab with monthly regimen	In 6.0 mg faricimab, 1.5 mg faricimab, and 0.3 mg ranibizumab mean improvements of +13.9, +11.7, and +10.3 ETDRS letters from baseline, respectively. The 6.0-mg faricimab dose showed statistically significant gain of +3.6 letters over ranibizumab (*p* = 0.03)	Completed
RHINE NCT03622593 [38]	III	DME	1715 patients; faricimab 6.0 mg every 8 weeks, faricimab 6.0 mg per PTI, or aflibercept 2.0 mg every 8 weeks	Faricimab every 8 weeks +11.8 ETDRS letters [10.6 to 13.0] vs. +10.3 ETDRS aflibercept 2.0 mg every 8 weeks letters [9.1 to 11.4] letters, difference +1.5 ETDRS letters [−0.1 to 3.2] Faricimab PTI +10.8 ETDRS letters [9.6 to 11.9], difference +0.5 ETDRS letters [−1.1 to 2.1]	Completed
YOSEMITE NCT03622580 [38]	III	DME	1532 patients; faricimab 6.0 mg every 8 weeks, faricimab 6.0 mg per PTI, or aflibercept 2.0 mg every 8 weeks	Faricimab every 8 weeks + 10.7 ETDRS letters [97·52% CI 9.4 to 12.0] vs. +10.9 ETDRS letters with aflibercept 2.0 mg every 8 weeks letters [9.6 to 12.2], difference −0.2 ETDRS letters [−2.0 to 1.6] Faricimab PTI + 11.6 ETDRS letters [10.3 to 12.9], difference +0.7 ETDRS letters with 8-week aflibercept 2.0 mg [−1.1 to 2.5];	Completed
ELEVATUM NCT05224102 [43]	IV	DME	120 patients; faricimab 6.0 mg every 4 weeks up to week 20, followed by faricimab 6.0 mg once every 8 weeks up to week 52	Awaiting results	Recruiting
RHONE-X NCT04432831 [39]	III	DME	1479 patients; faricimab 6.0 mg PTI up to 16 weeks	Awaiting results	Active, not recruiting
BALATON NCT04740905 [41]	III	BRVO	553 patients; faricimab 6.0 mg Q4W (Part 1) followed by faricimab PTI (Part 2) vs. aflibercept 2.0 mg Q4W (Part 1) followed by faricimab PTI (Part 2)	Awaiting results	Active, not recruiting
COMINO NCT04740931 [42]	III	CRVO/HRVO	730 patients, faricimab 6.0 mg Q4W (Part 1) followed by faricimab PTI (Part 2) vs. aflibercept 2.0 mg Q4W (Part 1) followed by faricimab PTI (Part 2)	Awaiting results	Active, not recruiting

BCVA best-corrected visual acuity; BRVO branch retinal vein occlusion; CRVO central retinal vein occlusion; DME diabetic macular edema; HRVO hemiretinal vein occlusion; w-AMD wet age-related macular degeneration.

## Data Availability

No applicable.

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
