# Peer review of "Faricimab for the Treatment of Diabetic Macular Edema and Neovascular Age-Related Macular Degeneration"

_pharmaceutics, 2023, doi:10.3390/pharmaceutics15051413_

Round 1

Reviewer 1 Report (Previous Reviewer 4)

The resubmitted form of the manuscript responded to all previous queries. I recommend publication.

Author Response

We thank the reviewer for this comment

Reviewer 2 Report (New Reviewer)

Pharmaceutics

2305269

Faricimab for the treatment of diabetic macular edema and neo-2 vascular age-related macular degeneration 3

Lorenzo Ferro Desideri, Carlo Enrico Traverso, Massimo Nicolò, Marion Munk

The review proposes to present the results obtained in the different trials where faricimab was used alone or in comparison with other monoclonal drugs already approved for the management of w-AMD and DME. Targeting the angiopoietin (Ang)– tyrosine kinase endothelial receptors (Tie) pathway beyond the VEGF pathway may be a possible therapeutic strategy. The authors list the results of the phase I-III trials studies of faricimab.

Authors may insert a figure demonstrating the targets of faricimab obtained from in vivo studies in mice, including control of permeability and edema, neovascularization, neuronal loss, leukocyte recruitment to the retina, loss of retinal epithelial barrier closure, retinal vascular occlusions and fibrosis/retinal thickness.

Although it has been approved by regulatory agencies, the faricimab has adverse effects on patients. Authors may insert a Table demonstrative of ocular and systemic adverse events (AEs) related in phase I-III trials.

The preclinical studies that demonstrated that the simultaneous double inhibition of Ang-2 and VEGF-A superior to the activity anti–VEGF-A or anti–Ang-2 alone were the basis for the development of the new bispecific antibody, faricimab for the treatment of w-AMD and DME.

However, the results of the comparative analysis of faricimab show similar or non-inferior responses to other monoclonal antibodies; rather than showing superior therapeutic responses.

Discussion of limitations should be included.

References 27 and 30 are the same.

Author Response

The review proposes to present the results obtained in the different trials where faricimab was used alone or in comparison with other monoclonal drugs already approved for the management of w-AMD and DME. Targeting the angiopoietin (Ang)– tyrosine kinase endothelial receptors (Tie) pathway beyond the VEGF pathway may be a possible therapeutic strategy. The authors list the results of the phase I-III trials studies of faricimab.

We thank the reviewer for this comment.

Authors may insert a figure demonstrating the targets of faricimab obtained from in vivo studies in mice, including control of permeability and edema, neovascularization, neuronal loss, leukocyte recruitment to the retina, loss of retinal epithelial barrier closure, retinal vascular occlusions and fibrosis/retinal thickness.

We thank the reviewer for this comment. We elaborated a figure as suggested, in which the biomolecular mechanism of action of faricimab is represented

Although it has been approved by regulatory agencies, the faricimab has adverse effects on patients. Authors may insert a Table demonstrative of ocular and systemic adverse events (AEs) related in phase I-III trials.

We thank the reviewer for this comment; however we deem that AEs are already well summarized in the paragraph 'safety' and an additive table would not add significant information in this regard for the reader.

The preclinical studies that demonstrated that the simultaneous double inhibition of Ang-2 and VEGF-A superior to the activity anti–VEGF-A or anti–Ang-2 alone were the basis for the development of the new bispecific antibody, faricimab for the treatment of w-AMD and DME. However, the results of the comparative analysis of faricimab show similar or non-inferior responses to other monoclonal antibodies; rather than showing superior therapeutic responses.

Discussion of limitations should be included.

We thank the reviewer for this comment.  A discussion of the suggested point has been included in the discussion section.

References 27 and 30 are the same.

We thank the reviewer for this comment. No they are different papers one published in 2016 and the other in 2017.

Reviewer 3 Report (New Reviewer)

The proposed review is about the use of Faricimab for the treatment of diabetic macular edema and neovascular age-related macular degeneration. The manuscript is well organized and suitable for the journal. However, a very important aspect regarding the safety of using the drug was not discussed.

Several paper describe the central retinal thickness fluctuations in patients treated with anti-VEGF drugs and this problem is discussed also for Faricimab. This aspect should be discussed and the suitable references should be added. For example:

- Sentaro Kusuhara et al., Short-Term Outcomes of Intravitreal Faricimab Injection for Diabetic Macular Edema, Medicina 2023, 59, 665.

- Ciucci et al., Central retinal thickness fluctuations in patients treated with anti-VEGF for neovascular age related macular degeneration, European Journal of Ophthalmology, 32, 4, 2388 - 2394.

- Stanga et al., Faricimab in neovascular AMD: first report of real-world outcomes in an independent retina clinic, Eye (Lond) 2023.

Author Response

The proposed review is about the use of Faricimab for the treatment of diabetic macular edema and neovascular age-related macular degeneration. The manuscript is well organized and suitable for the journal. However, a very important aspect regarding the safety of using the drug was not discussed.

Several paper describe the central retinal thickness fluctuations in patients treated with anti-VEGF drugs and this problem is discussed also for Faricimab. This aspect should be discussed and the suitable references should be added. For example:

- Sentaro Kusuhara et al., Short-Term Outcomes of Intravitreal Faricimab Injection for Diabetic Macular Edema, Medicina 2023, 59, 665.

- Ciucci et al., Central retinal thickness fluctuations in patients treated with anti-VEGF for neovascular age related macular degeneration, European Journal of Ophthalmology, 32, 4, 2388 - 2394.

- Stanga et al., Faricimab in neovascular AMD: first report of real-world outcomes in an independent retina clinic, Eye (Lond) 2023.

We thank the reviewer for this comment. We implemented this important point in the discussion section and added the suggested references.

This manuscript is a resubmission of an earlier submission. The following is a list of the peer review reports and author responses from that submission.

Round 1

Reviewer 1 Report

This manuscript by Desideri et al summarized the beneficial and side effects of administering Faricimab for the treatment of diabetic macular edema and neo-vascular age-related macular degeneration using evidences from animal models and clinical trials.  The content of this report is novel and it is suitable for publication with minor revision. I have some comments that need to be addressed first:

1.    Many drugs may have some gender, racial or ethical or age-related effects, I understand the numbers of patients enrolled in the clinical trials are relatively small, but the authors may give some discussion related to the possible gender, racial or ethical or age differences on guiding the usage of these drugs in the future.

2.    The authors may list all the abbreviations, it appears that some abbreviations don’t contain the full explanation of it, for example, FcyR, FA, CRT etc.

3.    The authors may list individual drugs’ half-life (T1/2). From this manuscript, I only know that Faricimab has a half-life of 7.5 days, how about other anti- DME and v-MAD drugs? Because the half-life of a drug will guide how long to use it.

4.    From line 106-107, “ In detail, in the Fc region FcyR was eliminated through the introduction of p329G mutations, which aims to shorten the systemic half-life (t1/2)”,  do you mean Faricimab have a short half-life after the Fc region modification?  Is there any special reason to shorten the half-life of Faricimab? Is it possible when performing intravitreal administration of Faricimab, longer half-life of a drug is better because it reduces the repetition frequency of the same drug. Please give a detailed explanation about it.

5.    There are some minor errors that need to be corrected.

a.       From line 24, please change “rapproved” to “approved”;

b.       From line 41, is it “ the consensual rise” or “ the consequential rise”?

c.       From line 125, “laser induced ? CNV” or “laser induced CNV”.

Thank you!

Author Response

  1. Many drugs may have some gender, racial or ethical or age-related effects, I understand the numbers of patients enrolled in the clinical trials are relatively small, but the authors may give some discussion related to the possible gender, racial or ethical or age differences on guiding the usage of these drugs in the future.
    We thank the reviewer for this comment. We discussed this point in the conclusion section.
  2. The authors may list all the abbreviations, it appears that some abbreviations don’t contain the full explanation of it, for example, FcyR, FA, CRT etc.
    We thank the reviewer for this comment. We modified the abbreviations as recommended by the reviewer.

  3. The authors may list individual drugs’ half-life (T1/2). From this manuscript, I only know that Faricimab has a half-life of 7.5 days, how about other anti- DME and v-MAD drugs? Because the half-life of a drug will guide how long to use it.

    We thank the reviewer for this comment. Serum t1/2 of other anti-VEGF agents has been described

  4. From line 106-107, “ In detail, in the Fc region FcyR was eliminated through the introduction of p329G mutations, which aims to shorten the systemic half-life (t1/2)”,  do you mean Faricimab have a short half-life after the Fc region modification?  Is there any special reason to shorten the half-life of Faricimab? Is it possible when performing intravitreal administration of Faricimab, longer half-life of a drug is better because it reduces the repetition frequency of the same drug. Please give a detailed explanation about it.
    We thank the reviewer for this comment. We better explained this point in the text (Please, see the highlighted part)
  5. There are some minor errors that need to be corrected.
    We thank the reviewer for this comment. We modified the suggested parts
  6. From line 24, please change “rapproved” to “approved”;
    We thank the reviewer for this comment. We modified this part.
  7. From line 41, is it “ the consensual rise” or “ the consequential rise”?
    We thank the reviewer for this comment. We modified this part in consequential rise.
  8. From line 125, “laser induced ? CNV” or “laser induced CNV”.
    We thank the reviewer for this comment. We modified this part in laser-induced CNV

Reviewer 2 Report

I would like to thank the authors for their comprehensive and timely paper. At this moment faricimab is introducing in many countries and this information will be useful for clinicians. I have only a few points to note:

i found no reference in the paragraph “PHASE I STUDIES”

“dose phase at day 1 and several intravitreal administrations on days 1, 28, and 56 in multiple-dose components.” sounds like the patients received two doses on day 1

I think it would be useful to indicate the number of patients maintaining Q16W or >Q12W regiments and compare them with those of aflibercept.

“Real-life studies have often provided inferior clinical outcomes 465 in comparison with large, multicenter pivotal trials (47, 48).” Please indicate the average difference and the dependence of the real-world outcomes from the number of injections

“of intraocular inflammation and/or retinal vascular occlusions accounting for 2.1% of the study population (49).” Please also provide the rate of vasculitis and/or RO (0.6%)

Author Response

I would like to thank the authors for their comprehensive and timely paper. At this moment faricimab is introducing in many countries and this information will be useful for clinicians. I have only a few points to note:
We thank the reviewer for this comment,

i found no reference in the paragraph “PHASE I STUDIES”
We thank the reviewer for this comment; however, the reference (34) is present in the above mentioned paragraph.

“dose phase at day 1 and several intravitreal administrations on days 1, 28, and 56 in multiple-dose components.” sounds like the patients received two doses on day 1
We thank the reviewer for this comment. Sentence was rephrased.

I think it would be useful to indicate the number of patients maintaining Q16W or >Q12W regiments and compare them with those of aflibercept.
We thank the reviewer for this comment. We implemented this analysis in the manuscript.

“Real-life studies have often provided inferior clinical outcomes 465 in comparison with large, multicenter pivotal trials (47, 48).” Please indicate the average difference and the dependence of the real-world outcomes from the number of injections. 
We thank the reviewer for this comment. We better discussed this poin in the discussion section providing the requested data.

Reviewer 3 Report

This paper could be resume with

Faricimab is a new bispecific antibody that targets both VEGF-A and the Ang-Tie pathway, which has the potential to solve some challenges in the treatment of macular exudative diseases such as wet age-related macular degeneration (w-AMD) and diabetic macular edema (DME). While intravitreal anti-VEGF drugs are currently the first-line therapy, they have limitations such as high treatment burden and unsatisfactory results in some patients. Faricimab has been approved by both the FDA and EMA for treating w-AMD and DME, with phase III trial results demonstrating its ability to maintain clinical efficacy with more prolonged treatment regimens compared to aflibercept (12 or 16 weeks) and a good safety profile.

Background

One potential criticism of the content is that it does not provide a comprehensive overview of the existing treatments for wet-aged related macular degeneration (w-AMD) and diabetic retinopathy (DR) with diabetic macular edema (DME). While the article mentions that intravitreal drugs targeting vascular endothelial growth factor (VEGF) are the current mainstay for treating these conditions, it does not provide a detailed comparison of the different treatment options or their respective efficacy and safety profiles.

Another potential criticism is that the article relies heavily on data from clinical trials sponsored by the manufacturer of faricimab, Roche/Genentech. Although the article mentions that real-life studies with anti-VEGF agents for treating w-AMD and DME have shown inferior results as compared with pivotal clinical trials, it does not provide a thorough analysis of the limitations and biases of these trials or of the potential conflicts of interest involved in their design and interpretation.

Furthermore, the article presents the angiopoietin (Ang)– tyrosine kinase endothelial receptors (Tie) pathway and the bispecific antibody faricimab as promising alternatives to anti-VEGF agents for the treatment of w-AMD and DME, but it does not provide a critical evaluation of the preclinical and clinical evidence supporting these claims. For instance, the article mentions that preclinical studies have highlighted the superiority of simultaneous dual inhibition of Ang-2 and VEGF-A over either anti–VEGF-A or anti–Ang-2 activity alone, but it does not provide a detailed analysis of the methodological limitations or potential confounders of these studies.

Finally, the article does not discuss the potential limitations and risks associated with the use of faricimab or other novel treatments for w-AMD and DME, such as the possibility of adverse effects or the need for frequent monitoring and follow-up. Overall, while the article provides a general overview of the prevalence and current treatments of w-AMD and DME, it would benefit from a more critical and balanced analysis of the available evidence and the potential implications of faricimab and other emerging treatments for these conditions.

Preclinical pharmacology

Line 97: The sentence is incomplete and lacks information about what "both" refers to.

Line 98: "This biological agent" is vague and could be specified to "Faricimab".

Line 99: The structure of Faricimab is not fully explained, as it is unclear what the "modified fragment crystallizable region" refers to.

Line 101: The comparison of Faricimab's molecular size to that of bevacizumab is not significant to the content of the paragraph and should be omitted.

Line 105: The information about brolucizumab is not relevant to the topic of Faricimab's structure and should be excluded.

Line 106-108: The reason behind eliminating FcyR and reducing viscosity is not explained, leaving the reader to wonder about the significance of these changes.

Line 110: "CrossMAB technology" is not fully explained and should be elaborated on.

Line 117: The sentence is incomplete and needs further information to be fully understood.

Line 123: The abbreviation "PDR" is not defined, and its meaning should be explained.

Line 127: The meaning behind "JR5558 mice models" is not clear and could be elaborated on.

Line 131: The sentence is incomplete and lacks information about what the inhibition of Ang-2 and VEGF-A induced.

Line 136: The sentence structure is unclear, and the meaning of "some anti-inflammatory properties" is not fully explained.

Line 139: The use of "exerted" is not necessary and could be omitted.

Line 142: The sentence structure is unclear, and the meaning of "diabetic mouse deficient in Ang-1" is not fully explained.

The content discusses the use of faricimab, a bispecific antibody that binds to both VEGF-A and Ang-2, for the treatment of exudative retinal disorders, including w-AMD and DME. The article cites clinical trials that demonstrate the prolonged efficacy of faricimab up to a Q16W treatment regimen, without being inferior to aflibercept in terms of visual and anatomical outcomes. The article also notes that faricimab has a good safety profile, with an incidence of ocular and systemic adverse events comparable to that of aflibercept. The article highlights the potential of faricimab to achieve stable and durable results and to reduce the treatment burden and costs for both healthcare providers and patients. The article provides a valid scientific criticism and supports the information presented in the article with relevant references. However, the content could have been more concise and structured.

Author Response

Faricimab is a new bispecific antibody that targets both VEGF-A and the Ang-Tie pathway, which has the potential to solve some challenges in the treatment of macular exudative diseases such as wet age-related macular degeneration (w-AMD) and diabetic macular edema (DME). While intravitreal anti-VEGF drugs are currently the first-line therapy, they have limitations such as high treatment burden and unsatisfactory results in some patients. Faricimab has been approved by both the FDA and EMA for treating w-AMD and DME, with phase III trial results demonstrating its ability to maintain clinical efficacy with more prolonged treatment regimens compared to aflibercept (12 or 16 weeks) and a good safety profile.

We thank the reviwer for this comment

Background

One potential criticism of the content is that it does not provide a comprehensive overview of the existing treatments for wet-aged related macular degeneration (w-AMD) and diabetic retinopathy (DR) with diabetic macular edema (DME). While the article mentions that intravitreal drugs targeting vascular endothelial growth factor (VEGF) are the current mainstay for treating these conditions, it does not provide a detailed comparison of the different treatment options or their respective efficacy and safety profiles.

We thank the reviewer for this comment. We added in the discussion a comparitive paragraph on the current approved anti-VEGF agents, dealing with both efficacy and safety.

Another potential criticism is that the article relies heavily on data from clinical trials sponsored by the manufacturer of faricimab, Roche/Genentech. Although the article mentions that real-life studies with anti-VEGF agents for treating w-AMD and DME have shown inferior results as compared with pivotal clinical trials, it does not provide a thorough analysis of the limitations and biases of these trials or of the potential conflicts of interest involved in their design and interpretation.

We thank the reviewer for this comment, we better discussed this topic in the discussion; it should also be taken into consideration that faricimab has only recently approved, and of course we need to wait for real-life studies to compare their results to those published by sponsored clinical trials.

Furthermore, the article presents the angiopoietin (Ang)– tyrosine kinase endothelial receptors (Tie) pathway and the bispecific antibody faricimab as promising alternatives to anti-VEGF agents for the treatment of w-AMD and DME, but it does not provide a critical evaluation of the preclinical and clinical evidence supporting these claims. For instance, the article mentions that preclinical studies have highlighted the superiority of simultaneous dual inhibition of Ang-2 and VEGF-A over either anti–VEGF-A or anti–Ang-2 activity alone, but it does not provide a detailed analysis of the methodological limitations or potential confounders of these studies.
We thank the reviewer for this comment. We implemented this part in the discussion

Finally, the article does not discuss the potential limitations and risks associated with the use of faricimab or other novel treatments for w-AMD and DME, such as the possibility of adverse effects or the need for frequent monitoring and follow-up. Overall, while the article provides a general overview of the prevalence and current treatments of w-AMD and DME, it would benefit from a more critical and balanced analysis of the available evidence and the potential implications of faricimab and other emerging treatments for these conditions.
We thank the reviewer for this comment. We modified the discussion.

Preclinical pharmacology

Line 97: The sentence is incomplete and lacks information about what "both" refers to.
We thank the reviewer for this comment. We modified it

Line 98: "This biological agent" is vague and could be specified to "Faricimab".
We thank the reviewer for this comment. We modified it

Line 99: The structure of Faricimab is not fully explained, as it is unclear what the "modified fragment crystallizable region" refers to.

Line 101: The comparison of Faricimab's molecular size to that of bevacizumab is not significant to the content of the paragraph and should be omitted.
We thank the reviewer for this comment. We omitted this part.

Line 105: The information about brolucizumab is not relevant to the topic of Faricimab's structure and should be excluded.
We thank the reviewer for this comment. We omitted this part.

Line 106-108: The reason behind eliminating FcyR and reducing viscosity is not explained, leaving the reader to wonder about the significance of these changes.
We better described in the preclinical pharmacology section

Line 110: "CrossMAB technology" is not fully explained and should be elaborated on.
We thank the reviewer for this comment. We better explained it

Line 117: The sentence is incomplete and needs further information to be fully understood.
We implemented this part in the text.

Line 123: The abbreviation "PDR" is not defined, and its meaning should be explained.
We thank the reviewer for this comment. We explained it

Line 127: The meaning behind "JR5558 mice models" is not clear and could be elaborated on.
We thank the reviewer for this comment. We modified it in the text

Line 131: The sentence is incomplete and lacks information about what the inhibition of Ang-2 and VEGF-A induced.
We thank the reviewer for this comment. We better explained this part.

Line 136: The sentence structure is unclear, and the meaning of "some anti-inflammatory properties" is not fully explained.
We thank the reviewer for this comment. we modified this par tin the text

Line 139: The use of "exerted" is not necessary and could be omitted.
We thank the reviewer for this comment. We omitted this term.

Line 142: The sentence structure is unclear, and the meaning of "diabetic mouse deficient in Ang-1" is not fully explained.

We thank the reviewer for this comment. We better explained it in the text

The content discusses the use of faricimab, a bispecific antibody that binds to both VEGF-A and Ang-2, for the treatment of exudative retinal disorders, including w-AMD and DME. The article cites clinical trials that demonstrate the prolonged efficacy of faricimab up to a Q16W treatment regimen, without being inferior to aflibercept in terms of visual and anatomical outcomes. The article also notes that faricimab has a good safety profile, with an incidence of ocular and systemic adverse events comparable to that of aflibercept. The article highlights the potential of faricimab to achieve stable and durable results and to reduce the treatment burden and costs for both healthcare providers and patients. The article provides a valid scientific criticism and supports the information presented in the article with relevant references. However, the content could have been more concise and structured.

Reviewer 4 Report

The article a a review on the use and outcomes of intravitreal Faricimab in the treatments of diabetic retinopathy and age related macular degeneration.

The subject is of interest, as the drug is relatively newly introduced in clinical practice.

However, the review needs to be better structured, so that it can be of scientific use for readers:

A short presentation of the databases used for research, strategy, inclusion and exclusion criteria for selected studies would be interesting for the reader.

The studies should be presented comparatively , rather than separately. The outcomes in terms of Visual acuity and macular thickness may be better understood when presented in comparative tables.

Sections such as Indications or Drug interactions should be removed, they look like as in a drug information sheet, not a scientific review.

However, side-effects should be discussed, and the safety of faricimab compared with other anti-VEGF agents.

Author Response

The article a a review on the use and outcomes of intravitreal Faricimab in the treatments of diabetic retinopathy and age related macular degeneration.

The subject is of interest, as the drug is relatively newly introduced in clinical practice.
We thank the reviewer for this comment

However, the review needs to be better structured, so that it can be of scientific use for readers:

A short presentation of the databases used for research, strategy, inclusion and exclusion criteria for selected studies would be interesting for the reader.

We thank the reviewer for this comment; we added a paragraph dealing with the methods of our research.

The studies should be presented comparatively , rather than separately. The outcomes in terms of Visual acuity and macular thickness may be better understood when presented in comparative tables.

We thank the reviewer for this comment; we added a comparative presentation of BCVA gains among the different trials in the table.

Sections such as Indications or Drug interactions should be removed, they look like as in a drug information sheet, not a scientific review.
We thank the reviewer for this comment. We removed that section from the text.

However, side-effects should be discussed, and the safety of faricimab compared with other anti-VEGF agents
We thank the reviewer for this comment; we added a paragraph in the discussion better discussion faricimab safety profile in comparison with other anti-VEGFs.